

# Soil moisture dynamics under two rainfall frequency treatments drive early spring CO₂ gas exchange of lichen-dominated biocrusts in central Spain

Selina Baldauf[1], Mónica Ladrón de Guevara[2], Fernando T. Maestre[2] and Britta Tietjen[1,3]

[1] Freie Universität Berlin, Department of Biodiversity/Theoretical Ecology, Institute of Biology, Berlin, Germany
[2] Departamento de Biología y Geología, Física y Química Inorgánica, Escuela Superior de Ciencias Experimentales y Tecnología, Universidad Rey Juan Carlos, Móstoles, Spain
[3] Berlin-Brandenburg Institute of Advanced Biodiversity Research (BBIB), Berlin, Germany

Corresponding author
Selina Baldauf,
selina.baldauf@fu-berlin.de

## ABSTRACT

**Background:** Biocrusts, communities dominated by mosses, lichens, cyanobacteria, and other microorganisms, largely affect the carbon cycle of drylands. As poikilohydric organisms, their activity time is often limited to short hydration events. The photosynthetic and respiratory response of biocrusts to hydration events is not only determined by the overall amount of available water, but also by the frequency and size of individual rainfall pulses.

**Methods:** We experimentally assessed the carbon exchange of a biocrust community dominated by the lichen *Diploschistes diacapsis* in central Spain. We compared the effect of two simulated precipitation patterns providing the same overall amount of water, but with different pulse sizes and frequency (high frequency: five mm/day vs. low frequency: 15 mm/3 days), on net/gross photosynthesis and dark respiration.

**Results:** Radiation and soil temperature, together with the watering treatment, affected the rates of net and gross photosynthesis, as well as dark respiration. On average, the low frequency treatment showed a 46% ± 3% (mean ± 1 SE) lower rate of net photosynthesis, a 13% ± 7% lower rate of dark respiration, and a 24% ± 8% lower rate of gross photosynthesis. However, on the days when samples of both treatments were watered, no differences between their carbon fluxes were observed. The carbon flux response of *D. diacapsis* was modulated by the environmental conditions and was particularly dependent on the antecedent soil moisture.

**Discussion:** In line with other studies, we found a synergetic effect of individual pulse size, frequency, environmental conditions, and antecedent moisture on the carbon exchange fluxes of biocrusts. However, most studies on this subject were conducted in summer and they obtained results different from ours, so we conclude that there is a need for long-term experiments of manipulated precipitation impacts on the carbon exchange of biocrusts. This will enable a more complete assessment of the impacts of climate change-induced alterations in precipitation patterns on biocrust communities.

## INTRODUCTION

Biocrusts are communities dominated by cyanobacteria, algae, fungi, lichens, and bryophytes living on the soil surface. They are a major feature in drylands worldwide, with an estimated cover of around 12% of the terrestrial surface (*Rodriguez-Caballero et al., 2018*), and act as a boundary layer between the soil and the atmosphere (*Belnap, Büdel & Lange, 2001*; *Belnap, Weber & Büdel, 2016*). Biocrust constituents, such as lichens and mosses, are poikilohydric organisms, which are dormant when desiccated and, unlike vascular plants, cannot actively control their water balance. Their activity time is thus limited to hydration events with water inputs from the atmosphere in the form of rain, fog, dew, water vapor, or melting snow (*Lange, 2001*; *Darrouzet-Nardi et al., 2015*). Despite their restricted activity time, biocrusts play an important role in the carbon cycle of drylands through their photosynthetic and respiratory activity (*Weber et al., 2012*; *Sancho et al., 2016*). The duration of moisture availability thereby controls the ratio of respiratory losses to photosynthetic gains (*Jeffries, Link & Klopatek, 1993*; *Belnap, Phillips & Miller, 2004*). Therefore, larger rainfall events are generally associated with higher carbon gains, whereas small rain events can result in a carbon deficit, because the initial respiratory losses cannot be compensated by subsequent photosynthesis (*Coe, Belnap & Sparks, 2012*; *Reed et al., 2012*; *Su, Lin & Zhang, 2012*).

In addition to the importance of individual event size, rainfall frequency is a key factor in determining the duration of moisture availability at the soil surface (*D'Odorico & Porporato, 2004*), which affects the activity of biocrust constituents (*Raggio et al., 2017*). As an example, a laboratory study with the common biocrust moss *Syntrichia caninervis* showed that a higher frequency of sufficiently large rainfall events is beneficial in terms of photosynthesis (*Coe, Belnap & Sparks, 2012*). These authors found that a shift in the interval between two wetting events from 1 day to 5 days led to a decrease in the mean carbon balance by nearly 90%, and became negative at an interval of 10 days. Furthermore, the effect of watering patterns on the carbon flux response depends on the season considered, as it plays an important role in determining the event size that is necessary to reach the compensation point for net photosynthesis (*Lange, 2001*; *Büdel et al., 2009*). Precipitation patterns can be beneficial in winter and lead to carbon losses in summer due to higher temperatures (*Darrouzet-Nardi et al., 2015*) and different physiological responses to rainfall events of the same size (*Coe, Belnap & Sparks, 2012*).

A spring to fall rainfall manipulation experiment on the Colorado Plateau (southwestern United States) showed that a 50% above-average precipitation frequency negatively impacts quantum yield, chlorophyll α, and protective pigment concentration in lichen- and cyanobacterial biocrusts, especially if applied for a longer period (*Belnap, Phillips & Miller, 2004*). In another study located in the same area, an increase in the frequency of small summer rainfall events led to a negative carbon balance and rapid mortality of *S. caninervis* (*Reed et al., 2012*; *Zelikova et al., 2012*). The additional small rainfall events caused C starvation due to non-sufficient hydration of the moss (*Reed et al., 2012*). In the same experiment, soil cyanobacterial biomass and abundance also declined significantly following the second summer of altered precipitation patterns
although the treatment increased overall moisture availability compared to control conditions (*Johnson et al., 2012*). In the long term, an increase in the summer precipitation frequency has been found to drive a shift in species composition from lichen- and moss-dominated to cyanobacteria-dominated biocrusts (*Zelikova et al., 2012*; *Ferrenberg, Reed & Belnap, 2015*).

Precipitation and water availability are the major driver of biocrust activity (*Wertin et al., 2012*) and hence, changes in precipitation patterns, such as those projected for climate change in drylands worldwide (*Pachauri et al., 2014*), will directly and indirectly influence the ability of biocrusts to fix and store carbon (*Belnap, Phillips & Miller, 2004*). However, we still know little about the impact of changes in precipitation on the carbon uptake across biocrust communities in different dryland regions and across seasons. In order to contribute to filling this knowledge gap, we have conducted a short-term manipulative experiment in early spring to assess the effect of rainfall patterns on photosynthesis and respiration of *Diploschistes diacapsis*, a lichen that generally is dominant in the biocrust communities in central Spain (*Maestre et al., 2011*). We compared the carbon exchange fluxes in response to two rainfall patterns that provided the same overall water amount; in one of the treatments, single rainfall events were smaller, but more frequent; while in the other one, single rainfall events were larger, but less frequent. By doing so, we seek to understand which rainfall pattern could be more beneficial for carbon fixation by biocrusts, and to set the direction for follow-up long-term experiments in the field.

## METHODS

### Sample collection and experimental design

In March 2017, we collected 12 undisturbed soil cores (diameter: 10 cm, height: seven cm) with a high crust cover (between 80% and 100%) of *D. diacapsis* at the rural estate El Espartal, in the Southeast Regional Park of Madrid, central Spain (40°11′N 3°36′W, 574 m a.s.l.). The bioclimate of El Espartal is upper semiarid meso-Mediterranean, with a mean annual temperature of 14.5 °C and a mean annual precipitation of around 389 mm in the period from 1971 to 2000 (*Cano Sánchez, 2006*). The mean precipitation in March and April is 36 and 52 mm, respectively. However, the historical minimum and maximum monthly precipitation (1981–2010) are 1 and 196 mm in March and 11 and 162 mm in April with maximum daily precipitation amounts of 29 mm in March and 45 mm in April (*Agencia Estatal De Meteorología, 2012*). The soils of the area are classified as Gypsic Regosols (*Ortíz-Bernard et al., 1997*) and Calcic Gypsisol (*Monturiol Rodriguez & Alcala Del Olmo, 1990*). Perennial plant coverage is lower than 40% and the site hosts a well-developed biocrust community dominated by lichens, such as *D. diacapsis*, *Squamarina lentigera*, *Fulgensia subbracteata*, and *Buellia zoharyi* (see Fig. S1).

After collection, we covered the bottom of the soil cores with a fine-meshed fabric to avoid soil loss and then took them to the lab, where they were watered to full saturation with low mineralized water and drained for 24 h to determine the weight at saturation water content. The undisturbed cores were placed on a structure that allowed water to drain from the cores under a transparent roof at the Climate Change Outdoor Laboratory,
located at the facilities of Rey Juan Carlos University (Móstoles, Spain: 40°20′N, 3°52′W, 650 m a.s.l., see Fig. S2). Rainfall was excluded by the roof, whereas temperature and radiation remained similar to those at ambient conditions. Five days after placing the cores under the roof, we started to apply a daily water pulse of 5 mm to six of the samples (high frequency treatment), and a 15 mm pulse at a 3-day interval to the other six samples (low frequency treatment). In total, all samples received a water amount of 60 mm during 12 days (from March 21st, 2017 until April 1st, 2017). Although *D. diacapsis* can activate the photosynthetic system at smaller rainfall pulses (*Lange et al., 1997*; *Pintado et al., 2005*), the watering patterns were chosen such that they were sufficient to provide enough water to both stimulate a pulse of lichen activity and wet the soil beneath the sample. Also, we wanted to ensure that the pulses exceed the threshold for ecologically effective precipitation, which has been reported to be five mm in a semiarid steppe ecosystem (*Hao et al., 2013*).

## Measurements

One of the six samples from each treatment was used to monitor soil temperature at three cm depth with a temperature sensor (UP Umweltanalytische Produkte GmbH, Cottbus, Germany). In the other five soil cores, we conducted gas exchange measurements using a Li-6400 portable photosynthesis system (Li-Cor, Lincoln, NE, USA). Measurements were taken every day after the water application and started earliest at 9:30 am. The mean time between water application and the first measurement of the samples was 77 min (SD = 13 min). In each sample, light and dark measurements were paired. First, the net $CO_2$ flux was measured by placing a transparent chamber on the sample (hereafter referred to as net photosynthesis), waiting for equilibrium to be reached between the chamber (see *Ladrón De Guevara et al., 2015* for a detailed description of the chamber) and the sample prior to measurement. For each measurement under light conditions, we recorded photosynthetically active radiation (PAR) with a Field Scout Quantum Light Meter (Spectrum Technologies, Plainfield, IL, USA). Next, measurements were conducted in the dark using the same technique as before, but this time placing an opaque cloth around the chamber to exclude light (hereafter referred to as dark respiration). Similar to the net photosynthesis measurements, we waited for the gas exchange to stabilize before taking the measurements, a process which took ca. 3–5 min. We assume, that after this time of dark-acclimation and stabilization, there is no longer a dynamic evolution of gas exchange due to a decrease in residual photosynthesis (compare, *Smith & Griffiths, 1996*, *1998*). Between each measurement, the infrared gas analyzers were standardized using the "match" procedure of the measuring device. With these constraints, the paired measurements were taken at the closest time possible to avoid a shift in the environmental conditions between them. The difference between the paired light and dark measurements was calculated, and will be referred to as gross photosynthesis hereafter. As the underlying soil was not removed from the crusts, the measurements comprise the fluxes from both biocrusts and underlying soil. With this, we follow an ecological approach, taking into account the contribution of the whole soil profile and avoiding any mechanical disturbance to the lichen (e.g., by thallus clipping) that could affect its functioning.

Paired measurements were taken alternatingly between the samples receiving a high and low frequency watering treatment to minimize the differences in environmental variables between the measurements. In total, three paired measurements were taken daily for every sample with a mean interval of 88 min between measurements. The last measurements were taken with a mean time of 4.5 h after the watering of that day (for detailed information on the mean times since the last watering see Table S1). On the 23rd of March, only one paired measurement was made for each sample due to snow and heavy wind being registered. Further measurements on that day were cancelled to avoid damage to the measuring device. After daily gas exchange measurements, the sample weights were determined to calculate the volumetric soil moisture. An average of on-site bulk density measurements of 1.03 g cm$^{-3}$ was used for all calculations.

The occurrence of dew could potentially influence the photosynthetic activity in our microcosms. Therefore, we estimated dewpoint temperature according to *Lawrence (2005)*. For the calculation, we used the relative air humidity and temperature recorded by the Li-6400 device and compared it to the lichen surface temperature which was recorded by the device as well (see Fig. S3 for relative humidity, dewpoint and lichen surface temperature during the measurements). Relative humidity did not exceed 76% and the dewpoint never was reached. Hence, we assume that dew did not confound the response of the microcosms to the applied watering treatments.

## Data analysis

We applied linear mixed effect models to evaluate the effect of the high and low watering frequency treatment and the environmental conditions on dark respiration, net and gross photosynthesis. We used the R statistical software (*R Core Team, 2018*) with the "nlme" package (*Pinheiro et al., 2018*) to conduct these analyses. Due to a high correlation between PAR, air and soil temperature ($T_{soil}$), we used PAR as a covariate in the models of net and gross photosynthesis, and $T_{soil}$ in the model of respiration. Because of problems with the measuring device, soil temperature data on the first day were only available for the low frequency treatment from 11 to 12 am. We used the average of these temperatures to fill the day's measurement gaps because soil temperature did not differ substantially between treatments. The watering treatment, PAR or $T_{soil}$ and their interaction were included as fixed factors in the models. We followed a protocol for model selection based on the AIC and maximum likelihood ratio tests to compare between models (*Zuur et al., 2009*). We considered the sample identification as a random intercept to account for the repeated measures, and we used a continuous autocorrelation structure of order 1 (CAR1 correlation structure) to correct the temporal autocorrelation within each sample. We monitored the AIC to choose the best-fitting model and checked model assumptions using QQ-plots of the residuals and a plot of the standardized residuals vs. fitted values. Due to the heteroscedasticity of the residuals, we allowed each day to have a different variance structure. Next, we sequentially excluded the fixed factors from the full model starting with the interaction and compared the models using a maximum likelihood ratio test. In this way, we selected a final model that only contained terms significant at the 5% level. The final model was refitted with restricted maximum likelihood estimators.

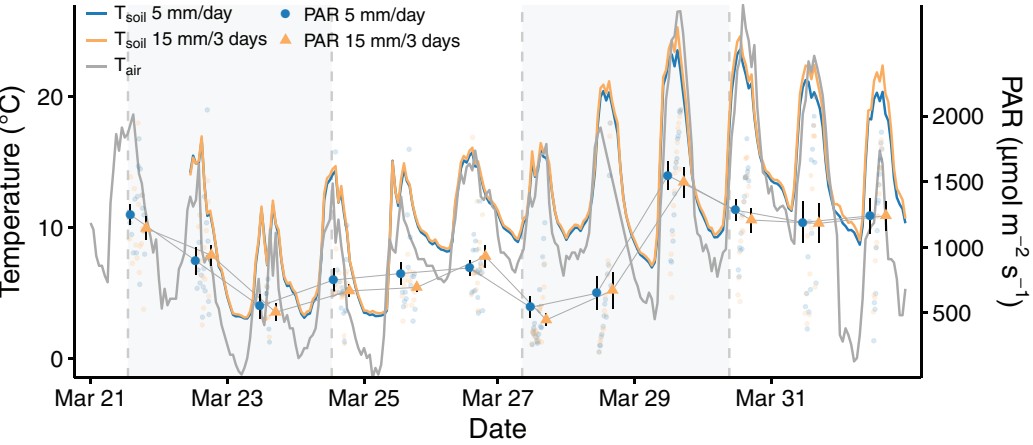

**Figure 1** **Time progression of PAR, soil and air temperature during the experiment.** Points and error bars represent the mean ± 1 SE ($n = 15$ for each data point in PAR), the smaller, transparent points are the individual PAR measurements. The dashed, gray vertical lines indicate the days on which samples of both treatments were watered and the gray background visually separates the measuring time into four periods of 3 days.                                

We performed Wilcoxon rank sum tests to compare the soil moisture of the two treatments on day 0, 1, and 2 since the last watering of the low frequency treatment. Because data were not normally distributed, paired Wilcoxon rank sum tests were used to compare $CO_2$ fluxes of dark respiration, net and gross photosynthesis between the two treatments overall and on the different days since the last watering. The same test was also used to compare soil temperatures between treatments during the measurement times. This data analysis was also performed using R statistical software (*R Core Team, 2018*).

# RESULTS

## Environmental conditions during the experiment

The temporal dynamics of the measured environmental variables (PAR, soil and air temperature and soil moisture) were highly variable during the experiment (see Figs. 1 and 2). PAR values ranged from 50 to 2,000 µmol m$^{-2}$ s$^{-1}$, and exceeded the light compensation point of *D. diacapsis* of around 100 µmol m$^{-2}$ s$^{-1}$ (*Lange, 2001*) for all but five measurements. Ambient air and soil temperatures ranged from 6 to 31 °C and from 3 to 25 °C, respectively. These wide ranges reflect the variable climatic conditions during the experimental phase, with PAR, air and soil temperature decreasing after March 21st and increasing towards the end of the experimental phase. Soil temperature was slightly lowered in the high frequency treatment samples ($W = 1,156$, $p < 0.001$). The difference in soil temperature was markedly higher during the last 4 days of the experiment (mean difference of 1 °C) compared to the period before (mean difference of 0.3 °C).

Soil moisture dynamics generally reflected the applied watering patterns. The samples of the low frequency treatment showed a decrease in soil moisture following the days after the 15 mm pulse. The soil moisture of the high frequency treatment increased during the first days because temperature and therefore evaporation were low and thus moisture accumulated in the samples. When air temperature exceeded 20 °C, evaporation was

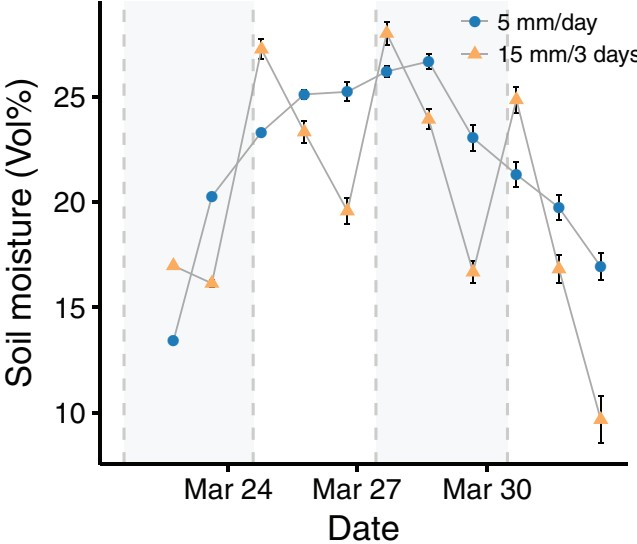

**Figure 2 Soil moisture dynamics during the experiment.** Points and error bars represent the mean ± 1 SE ($n$ = 5 for each data point). The dashed, gray vertical lines indicate the days on which samples of both treatments were watered and the gray background visually separates the measuring time into four periods of 3 days.

high enough for soil moisture to decrease. On the days when the samples of both treatments were watered, soil moisture was 13% higher for the low watering frequency samples, which received 15 mm ($W$ = 195, $p$ < 0.001). On the following day there was no significant difference in soil moisture between the treatments ($W$ = 246, $p$ = 0.221), and on the second day after watering, soil moisture was 27% lower for the low watering frequency samples ($W$ = 354, $p$ < 0.001). An overall comparison of soil moisture for the entire experimental period did not indicate moisture differences between the treatments ($W$ = 1,262, $p$ = 0.135). However, cumulative moisture was higher in the high frequency treatment for all four periods of 3 days (see Table S2 for details).

### Watering effects on $CO_2$ fluxes

Similar to the environmental conditions, the $CO_2$ flux measurements showed a high variability. On the first day of the experiment, we observed a negative net photosynthesis associated with high respiration rates in all samples measured (see Fig. 3A). On the second day, net photosynthesis increased and stayed above or close to zero for the remaining experiment. The flux difference between treatments became more pronounced from the 29th of March until the end of the experiment. During these days, temperature and radiation were higher and the largest flux differences were associated with the largest differences in soil moisture between treatments.

Overall, we observed a net $CO_2$ uptake (i.e., positive net photosynthesis) with mean net fluxes of 0.35 ± 0.05 (mean ± 1 SE) and 0.56 ± 0.06 µmol m$^{-2}$ s$^{-1}$ for the low and the high frequency treatment, respectively. When comparing the mean $CO_2$ fluxes of the two frequency treatments, all fluxes were significantly lower in the low frequency treatment, with 13 ± 7% (mean ± 1 SE) lower dark respiration ($W$ = 4,046, $p$ < 0.001),

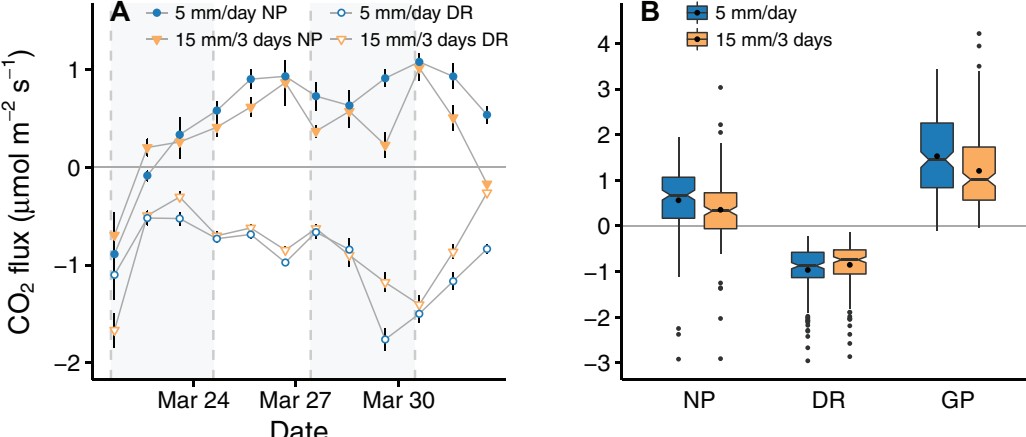

**Figure 3 Time progression (A) and overall comparison (B) of carbon flux measurements for the low and high frequency watering treatment.** Points and error bars (A) represent mean ± 1 SE ($n = 15$ for each data point) for net photosynthesis (closed symbols) and dark respiration (open symbols). The boxplots (B) of net photosynthesis (NP), dark respiration (DR), and gross photosynthesis (GP) include a black dot indicating the group mean ($n = 170$ in each group). The non-overlapping notches represent the 95% confidence interval of the median and suggest differences between treatments over the experimental period.                               

**Table 1 Median and 95% confidence interval of the median (in brackets) of the pairwise relative differences (%) between the two treatments by days since the last watering of the low frequency treatment.**

| Days since last watering of low frequency treatment | Net photosynthesis | Dark respiration | Gross photosynthesis |
|---|---|---|---|
| 0 | 6 (−30.68) | 1 (−4.8) | −2 (−14.20) |
| 1 | 15 (−44.64) | **7*** (−2.14) | 5 (−3.27) |
| 2 | **136**\*** (−92.182) | **38**\*** (27.52) | **71**\*** (55.91) |

Note:
  Significant differences resulting from a Wilcoxon rank sum test are marked in bold (\*$p < 0.05$, \*\*\*$p < 0.001$).

46 ± 3% lower net photosynthesis ($W = 9{,}818$, $p < 0.001$) and 24% ± 8% lower gross photosynthesis ($W = 17{,}602$, $p < 0.001$) (see Fig. 3B).

The relative difference between treatments changed with the days since the last watering of the low frequency treatment (see Table 1 and Fig. S4). On days when both treatments were watered, no significant flux differences between them were found (see Figs. 4A–4C), net photosynthesis: $W = 1{,}032$, $p = 0.391$; dark respiration: $W = 980$, $p = 0.635$; gross photosynthesis: $W = 931$, $p = 0.909$). In the low frequency treatment, the $CO_2$ fluxes for dark respiration, net and gross photosynthesis decreased on the days following water application. For dark respiration the difference between the two treatments was significant on the first and second day after watering ($W = 587$, $p = 0.016$). Net and gross photosynthesis were more variable among samples, therefore means only differed significantly on the second day following water application (net photosynthesis: $W = 1{,}034$, $p < 0.001$, gross photosynthesis: $W = 1{,}118$, $p < 0.001$).

The linear mixed model analysis (see Table S3 for results of final model) showed a significant influence of both the watering treatment and the environmental conditions on

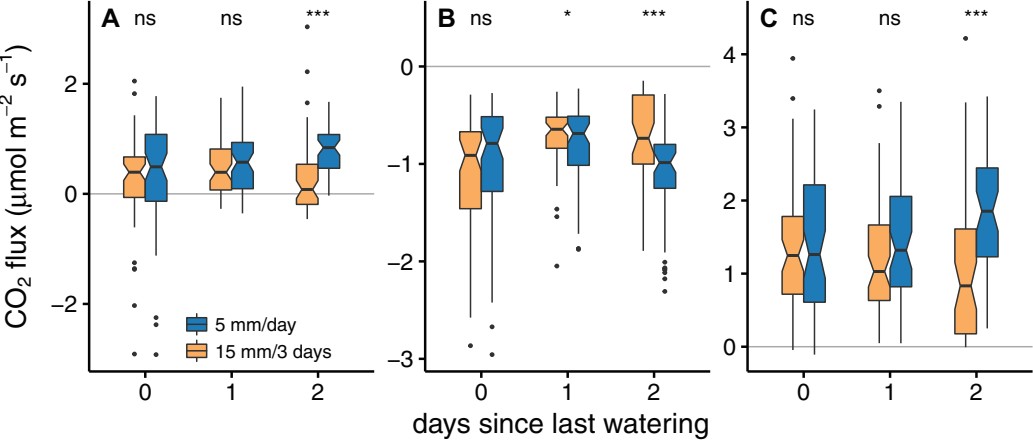

**Figure 4 Comparison of net photosynthesis (A), dark respiration (B), and gross photosynthesis (C) between the two treatments on the different days since the last watering of the low frequency treatment.** Measurements are displayed separately for each of the three low frequency interval days and compared to the corresponding values of the daily watered high frequency treatment. The asterisks indicate significant differences between the two treatments, calculated with a paired Wilcoxon rank sum test ($^*p < 0.05$; $^{***}p < 0.001$; ns, not significant, $n = 60$ for 0 and 1 days since the last watering and $n = 50$ for 2 days since the last watering). Note the significant difference in dark respiration on day 1 since the last watering despite largely overlapping notches, which arises due to the paired nature of the test. An unpaired test did not show significant differences in this case.

the measured carbon fluxes. We found a significant watering treatment:PAR interaction and a watering treatment:Tsoil interaction on net photosynthesis and on dark respiration, respectively (saturated model vs. model without interaction: d$f$ = 17, $L$-ratio = 6.96, $p$ = 0.008 for net photosynthesis and d$f$ = 17, $L$-ratio = 7.85, $p$ = 0.005 for dark respiration). Therefore, we could not exclude the single effects from the model. In contrast, the interaction between these variables was not found to be significant for gross photosynthesis (saturated model vs. model without interaction: d$f$ = 17, $L$-ratio = 2.65, $p$ = 0.104). The additive fixed terms PAR and the watering treatment both significantly affected the model (additive model vs. model without PAR: d$f$ = 16, $L$-ratio = 147.7, $p$ < 0.001, additive model vs. model without the watering treatment: d$f$ = 16. $L$-ratio = 4.2, $p$ = 0.040).

## DISCUSSION

The carbon balance of biocrust communities depends on the patterns of moisture availability, which are likely to be altered by climate change. Therefore, it is important to assess how different biocrust communities respond to an alteration in the rainfall event size and frequency and how regional and seasonal differences can influence this response. In this study, we assessed how two different precipitation frequency patterns providing the same overall amount of water (i.e. 5 mm/day vs. 15 mm/3 days) affected photosynthesis and respiration of biocrusts dominated by *D. diacapsis* in central Spain. We observed that radiation and soil temperature, together with the watering treatment, affected the rates of net and gross photosynthesis, as well as dark respiration.

During most of the experimental period, temperature was below the optimum of 22–24 °C for NP reported from *D. diacapsis* in Spain (*Pintado et al., 2005*). We observed the

highest carbon flux rates on the days when temperatures exceeded 20 °C. However, the rates of 4–5 µmol m$^{-2}$ s$^{-1}$ that are reported at optimum temperature (*Pintado et al., 2005*) were not reached in our experiment, where the maximum rates of net photosynthesis and dark respiration were ca. 3 µmol m$^{-2}$ s$^{-1}$. A major contribution to the difference of our measurements to the maximum net photosynthesis at optimum conditions reported in the literature is possibly the hydration status of the lichens, because PAR values exceed NP compensation point of *D. diacapsis* at this temperature (*Pintado et al., 2005*).

In addition to the different water availability in the treatments, the lowered soil temperature in the high frequency samples (0.3–1 °C difference) possibly contributed to the observed differences between the treatments due to the exponential sensitivity of respiration to temperature. A difference of 1 °C in temperature can lead to a difference of around 5–8% in additive respiration of bare soil and *D. diacapsis* at the temperatures relevant to our study (*Lange et al., 1997*; *Pintado et al., 2005*; *Castillo-Monroy et al., 2011*, see Fig. S5). However, these temperature differences can only explain a small part of the observed total changes in dark respiration and net photosynthesis. Thus, we conclude that the main driver of the differences is water availability but that soil temperature differences also contributed to the observed patterns.

## Individual pulse sizes

We did not find differences in the mean photosynthetic and respiratory response between the two pulse sizes evaluated (5 and 15 mm) on days when they were both applied. However, on the first day of the experiment, when soils and lichens were dry prior to watering, respiration, and net photosynthesis were higher in the samples receiving the larger pulse. The lichen *D. diacapsis* can achieve its maximum net photosynthesis rate at thallus water contents of around 0.5–0.75 mm precipitation equivalent (*Lange, 2001*; *Pintado et al., 2005*). There are studies showing an effect of suprasaturation at a relative water content of around 50% of the maximum water holding capacity (*Pintado et al., 2005*) and others that do not show a depression of photosynthesis at suprasaturation (*Lange, 2001*). Following these studies, we can assume that in our experiment both pulses were large enough for moisture within the lichen to exceed this threshold. However, it is unclear whether suprasaturation limited photosynthesis. An indication for a suprasaturation effect could be the fact that the larger watering pulse did not lead to a higher NP response, as we would have expected. Generally, larger rainfall events lead to longer periods of moist conditions and therefore, more carbon can be fixed (*Coe, Belnap & Sparks, 2012*; *Reed et al., 2012*). *Su, Lin & Zhang (2012)* reported higher net carbon fluxes and respiration rates from cyanobacterial-lichen crusted soils in the Gurbantunggut Desert after applying a single rainfall pulse of 15 mm compared to a 5 mm pulse applied on dry soil. On the first days of our study, the 15 mm pulse led to higher net photosynthesis despite the higher respiration compared to the 5 mm pulse. The rewetting of dry biocrust communities and the underlying soil can lead to large immediate carbon losses caused by physical processes, and increased respiration due to cell reparation processes (*Farrar, 1973*; *Smith & Molesworth, 1973*; *Lange, 2001*).

This explains the high respiration rates on the first day despite soil moisture being lower than in the following days. The beneficial effect of the 15 mm watering pulse was not sustained throughout the experiment, despite the soil moisture level after the water application being always higher for this pulse size. This indicates that not only the moisture content itself, but also the condition of a sample prior to moistening plays a role in the carbon exchange response of biocrusts. Applying the larger pulse on previously wetted soil was not beneficial in comparison to a smaller pulse that was already large enough to sufficiently wet the sample to trigger photosynthetic activity.

## Inter-pulse frequency

When looking at the simulated rainfall patterns, we found that smaller, more frequent watering pulses were beneficial in terms of net photosynthesis. The observed differences between treatments increased with differences in soil moisture, and were higher during the last days of the experiment (when temperature and radiation increased). Our findings suggest a synergetic effect between the individual watering pulse size, the frequency of its occurrence, and other environmental factors such as antecedent moisture, temperature, and radiation. Apart from the size and frequency of rainfall events, temperature is an important control for moisture availability in drylands (*Tietjen et al., 2017*) and evaporative losses increase with it. Consequently, the response of biocrusts to altered precipitation frequencies is seasonally different because the water amount necessary to exceed the photosynthetic net compensation point differs between summer and winter (*Lange, 2001*; *Büdel et al., 2009*). There is evidence that, apart from the indirect effect of temperature on water availability, desiccation tolerance is also a seasonally dynamic physiological property in lichens and mosses (*Green, Sancho & Pintado, 2011*).
For example, the carbon balance of the biocrust moss *S. caninervis* was higher for the same watering event when collected in winter compared to summer despite identical laboratory conditions (*Coe, Belnap & Sparks, 2012*).

The interacting effect of season and temperature with alterations in rainfall patterns on the performance of biocrust constituents is a common observation in many studies. The correlation between rainfall frequency and biocrust growth was found to be positive for winter rain and negative for summer rain areas in a study across southern African sites (*Büdel et al., 2009*). In the southwestern United States, the carbon balance of *S. caninervis* in response to rainfall frequency and amount was modulated by season and events leading to a positive carbon balance in winter could result in carbon losses in summer (*Coe, Belnap & Sparks, 2012*). Other studies from the same area reported a negative effect of an increased summer precipitation frequency on different variables related to the photosynthetic performance of cyanobacterial- and lichen-dominated soil crusts (*Belnap, Phillips & Miller, 2004*; *Johnson et al., 2012*; *Zelikova et al., 2012*).

To our knowledge, there are no rainfall manipulation studies assessing the response of *D. diacapsis* dominated biocrusts to different rainfall frequencies. Most of the studies evaluating how alterations in the precipitation frequency affect physiology and functioning of biocrust constituents have been conducted in summer, and therefore reported a negative impact of a higher frequency of small pulses. We conducted this study in early

spring, with rather cold nights and maximum temperatures around 30 °C. These moderate temperatures enabled a constant activation of lichen photosynthesis during daily measurements. The rainfall patterns applied in this study provided a total water amount of 60 mm, which is well above the monthly saturation rainfall of 40 mm that is reported for maximum biocrust activity across Europe (*Raggio et al., 2017*). In combination, this led to a positive net photosynthesis and crust activity during the experiment. Field measurements with lichen-dominated biocrusts in Spain show that periods of net carbon fixation occur during the winter months, and are commonly restricted to a few hours in the morning or the late afternoon; in summer and during midday, net photosynthesis is mostly negative (*Maestre et al., 2013*; *Ladrón De Guevara et al., 2014*; *Raggio et al., 2014*). In summer, potential evaporation rates are high and both biocrusts and the underlying soil dry out relatively quickly upon rewetting. For an identical experiment conducted in the summer time, we would therefore expect results that are more similar to those reported from the summer studies mentioned above. However, since most rainfall events in central Spain occur during the spring and autumn/winter period (*Lafuente et al., 2018*), the impact of different rainfall patterns should also be studied during these phases of high biocrust activity.

## CONCLUSION

In accordance with other studies, we showed that precipitation frequency plays an important role for the carbon balance of lichen-dominated biocrusts in central Spain. Previous studies revealed a large variability in the carbon exchange response of different biocrust constituents and communities to rainfall. To our knowledge, no studies so far have assessed the responses of *D. diacapsis* dominated biocrusts to altered early spring precipitation frequency. In contrast to what has been found in previous studies at high summer temperatures and conducted with other biocrust-forming lichens and mosses, our results indicate that at moderate temperatures, a higher rainfall frequency is beneficial given the same overall water amount over a short period. This clearly shows that the gas exchange response to different rainfall frequencies is modulated by radiation and temperature conditions leading to seasonal differences. We therefore call for detailed cross-site and cross-season comparisons of the impacts of altered rainfall patterns on the carbon fluxes of biocrust-dominated soils.

## ACKNOWLEDGEMENTS

The authors would like to thank Victoria Ochoa and Beatriz Gozalo for their assistance with the laboratory work and Arantzazu López de Luzuriaga for providing the space for the experiment.

### Funding

This research was supported by the Collaborative Research Centre 973 (www.sfb.973) of the German Research Foundation (DFG) and the European Research Council

(BIODESERT project, ERC Agreement No. 647038). The funders had no role in study design, data collection and analysis, decision to publish, or preparation of the manuscript.

## Grant Disclosures
The following grant information was disclosed by the authors:
Collaborative Research Centre 973 of the German Research Foundation (DFG) and the European Research Council: BIODESERT project, ERC Agreement No. 647038.

## Competing Interests
The authors declare that they have no competing interests.

## Author Contributions
- Selina Baldauf conceived and designed the experiments, performed the experiments, analyzed the data, prepared figures and/or tables, authored or reviewed drafts of the paper, approved the final draft.
- Mónica Ladrón de Guevara conceived and designed the experiments, performed the experiments, contributed reagents/materials/analysis tools, authored or reviewed drafts of the paper, approved the final draft.
- Fernando T. Maestre conceived and designed the experiments, contributed reagents/materials/analysis tools, authored or reviewed drafts of the paper, approved the final draft.
- Britta Tietjen conceived and designed the experiments, authored or reviewed drafts of the paper, approved the final draft.

## Data Availbility
Baldauf, Selina (2018): Data and R script: Soil moisture dynamics under two rainfall frequency treatments drive early spring $CO_2$ gas exchange of lichen-dominated biocrusts in central Spain. figshare. Collection. https://doi.org/10.6084/m9.figshare.c.4077107.v2.

## Supplemental Information
Supplemental information for this article can be found online at http://dx.doi.org/10.7717/peerj.5904#supplemental-information.

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
