# Peer review of "Soil moisture dynamics under two rainfall frequency treatments drive early spring CO2 gas exchange of lichen-dominated biocrusts in central Spain"

_PeerJ, doi:10.7717/peerj.5904_

## Round 0.1 · original submission · Major Revisions

Thank you for submitting your manuscript to PeerJ. The reviewers felt that the manuscript was well written, clearly conveying the primary findings of the research – the combinatorial effects of precipitation, temperature and PAR on C flux in biocrusts. The integration of the results into the existing literature was another strength of the manuscript. However, the reviewers also provide several insightful suggestions that would significantly improve the quality of the manuscript.

In particular, both reviewers were concerned that the existing figures do not illustrate the significant differences between the watering treatments as expressed in the text. They provide suggestions to address this discrepancy, and in one instance, suggest reanalyzing the data to verify the accuracy of the statistical information in the text.

Reviewer #1 questions several of the experimental approaches (how dark respiration was measured and the logistics of water treatments and subsequent measurements) and offers some relatively simple questions and tests to address the concerns.

Finally, it appears that increased soil moisture was the primary driver supporting net positive CO2 fixation under the relatively mild environmental conditions during which the experiments were conducted. There are a myriad of different precipitation patterns and other microclimatic conditions that could affect soil moisture, thus the title of the manuscript is somewhat misleading. If the precipitation amount and/or wetting frequency had been altered would the results have been the same?

Please address each of the reviewer’s discerning comments, with particular emphasis on the points raised above.

We look forward to your revised manuscript.

Reviewer 1 ·

Basic reporting

The article generally reflects the existing literature, is written clearly, and has nicely designed graphics. The hypotheses and results are conveyed clearly.

Experimental design

The experiment is well-designed and neatly circumscribed. The research questions are interesting and provide a moderately novel insight into biocrust lichen CO2 fluxes. With two caveats (described below) the investigation is carried out with sufficient technical proficiency. The methods would be relatively straightforward to replicate.

My first methodological concern regards how the dark respiration was measured. I have serious reservations about considering these measurements 'dark' measurements. In my experience, there is a 'dark-acclimation' period of at least several minutes prior to which biocrust carbon fixation appears to continue. I would appreciate if the authors are able to justify the immediate measurement of CO2 flux (with data) or conduct the very simple assay of measuring CO2 fluxes on a diacapsis specimen at different intervals after imposing darkness, for instance, 0,2,5,10 15 and 30 minutes afterward.

My second methodological concern is that the order of the watering treatment and the measurements may actually skew the results toward the desired conclusion. If the authors had measured 5 hours after the watering treatments, or anything other than 1 hour after the watering treatments, is it possible that the results would have been different? This is a speculative question, but if I understand correctly, the watering is done from the top, ensuring that the biocrust lichen was definitely wet immediately after watering. However, the lichen may have been equally dry, or even drier, several hours later, and fluxes may have been suppressed. This consideration might moderate the conclusions presented, and I think somewhat changes the effective experimental design. Can you somehow verify that the 1 day watering treatment actually stayed wetter than the 3 day treatment?

Validity of the findings

Generally the data and statistical analysis appear to be sound. However, the statistics and figures 2 and 3 don't visually correspond very well, and I am concerned that the statistical analysis is not conveyed accurately. Specifically, on lines 198-201 you describe significant differences in the text which really are not apparent from figure 2b. For the most part the intervals seem to overlap pretty substantially, and even the means seem quite close. Can you please verify that you are indeed comparing the values across all dates and times since watering? Additionally, in figure 3, middle panel, I am skeptical that the statistics here represent the same data and analysis as is displayed in the graph. It is hard to imagine how the dark respiration one day after watering is actually different between the two plots. the intervals entirely overlap, and the means are very close.

Reviewer 2 ·

Basic reporting

Please see comments for the author

Experimental design

Please see comments for the author

Validity of the findings

Please see comments for the author

Additional comments

This is a well-written research article presenting interesting information about the physiology of a representative lichen of biocrusts. The study of the effect of factors determining biological activity in drylands, such as rainfall patterns, is of the utmost importance, since predictions of climate change include altered precipitation regimes in addition of increasing temperatures. However, I have a few comments that I would recommend the authors to address:

1. Mean annual precipitation value is stated in methods, but since seasonal variations would be important in the effect of the physiological activities studied, Spring values should be reported, or at least explain why a water amount of 60 mm during the twelve days of study was selected, and/or why you study the differential effect of water pulses of 5 mm/day and 15 mm pulse at three-day interval, in order to know if these are representative of the environmental conditions at the studied site.

2. Evidence has shown that relative air humidity and dew could be an important water input in biocrust, with higher photosynthetic activity early in the morning than in other periods of the day, so timing of measurements should be reported and the possible effect of water vapour adsorption by biocrusts discussed.

3. Although it was specified in methods that the underlying soil was not removed to avoid disturbances, and that, therefore, measurements comprise the fluxes from both biocrusts and underlying soil, I would suggest for further experiments to add other samples in which the thallus of the lichen were removed in order to know if the CO2 exchange measurements are influenced by metabolic activity of bacteria or other organisms living in the adherent soil, or even “control” samples with sterilized soils.

4. Environmental conditions during the experiments are shown in separate Figures, and one of them (temperature conditions) included in Supplementary Fig. S3, so it is difficult to compare data, which are correlated, and it would be convenient to see in parallel. I suggest to combine both data in a figure, which could include in separate axes the different variables, as found in many papers.
In addition, regarding Figure 1, I cannot see values of 50 μmol m−2 s−1 or 2000 μmol m−2 s−1, as stated in the ranges described in the results section, please check it.

5. I was very surprised when I read that there were significant differences between treatments, because a first look of Figure 2A seems to show similar results between treatments, with almost overlapping symbols. It may be that the range of measurements on the axis is too extended, from 0 to 4 μmol m−2 s−1, since your values are close to zero, so I would suggest to change the scale on the axis in order to see the symbols better; for instance to represent data from 0 to 2 μmol m−2 s−1.

6. Evaluation of the results comparing previous results on other lichens and/or other geographic distant places has been discussed. However I found previous reports on Diploschistes diacapsis from biocrusts of another location in Spain (Pintado et al. ), in which photosynthetic activity was also studied regarding microclimatic conditions, that I think must be added and discussed.

---

## Round 0.2 · Minor Revisions

Dear Ms. Baldauf,

Thank you for the revised manuscript and clear, detailed responses to the reviewer's comments. Overall, the reviewers were satisfied with the changes to the manuscript (and your responses) and felt the paper should be a welcome addition to the literature. Reviewer 1 did, however, provide several suggestions that would benefit the manuscript. Once these minor suggestions are addressed please resubmit the manuscript for a quick decision. Also several grammatical suggestions:

Line 120: delete “already”
Line 143: replace “reminiscent” with “residual”.
Line 200: Should also reference Fig. 2 here since soil moisture is discussed.

Best regards,

Chris Yeager

Reviewer 1 ·

Basic reporting

No comment

Experimental design

My earlier critique of the experimental design has been adequately addressed.

Validity of the findings

No comment.

Additional comments

Line 48: If you really want to list the water inputs, listing snow is important.

Line 66: You might consider citing Darrouzet-Nardi, A., Reed, S.C., Grote, E.E. et al. Biogeochemistry (2015) 126: 363. https://doi.org/10.1007/s10533-015-0163-7 here. (I am not an author of this study.)

Figure 1: This is just an observation, but the authors might consider exploring it: It may be worth considering that the daily watering appears to have slightly lowered the maximum daily temperature during the hotter days. Respiration is exponentially temperature sensitive, while photosynthesis is not, such that a small difference on warmer days could lead to an outsized carbon imbalance. This might be an alternative pathway by which precip frequency is affecting CO2 fluxes in this system.

Figure 4: The order of the boxes (gold, blue) should not be reversed compared to the other figures (blue, gold). Further, panels should be in the order A, B, C.

Figure 4/line 238-246: This comment is in reference to the significant difference reported in figure 4 a, day 1. Since the statistics represent pairwise differences, but the boxplots are group-level differences, I think reporting in the text the magnitude of the pairwise differences is important. I have a hard time believing that this 'significant' result is substantial, and would like a quantitative estimate of the difference. In the end, I don't think it changes much about the interpretation, but as a minor statistical detail it is important.

Reviewer 2 ·

Basic reporting

After re-review this manuscript, I have been satisfied with the answers and use of the recommendations I made in my first review

Experimental design

No comment

Validity of the findings

No comment

Additional comments

After re-review this manuscript, I have been satisfied with the answers and use of the recommendations I made in my first review

---

## Round 0.3 · accepted · Accept

Thank you for the revised manuscript and the well thought out response to the reviewer's comments. The study will be a nice addition to biocrust literature.

#